# Distilling causality between physical activity traits and obesity via Mendelian randomization

Zhe Wang [1✉], George Davey Smith [2,3], Ruth J. F. Loos [1,4] & Marcel den Hoed [5✉]

## Abstract

**Background** Whether obesity is a cause or consequence of low physical activity levels and more sedentary time has not yet been fully elucidated. Better instrumental variables and a more thorough consideration of potential confounding variables that may influence the causal inference between physical activity and obesity are needed.

**Methods** Leveraging results from our recent genome-wide association study for leisure time moderate-to-vigorous intensity (MV) physical activity and screen time, we here disentangle the causal relationships between physical activity, sedentary behavior, education—defined by years of schooling—and body mass index (BMI), using multiple univariable and multivariable Mendelian Randomization (MR) approaches.

**Results** Univariable MR analyses suggest bidirectional causal effects of physical activity and sedentary behavior with BMI. However, multivariable MR analyses that take years of schooling into account suggest that more MV physical activity causes a lower BMI, and a higher BMI causes more screen time, but not vice versa. In addition, more years of schooling causes higher levels of MV physical activity, less screen time, and lower BMI.

**Conclusions** In conclusion, our results highlight the beneficial effect of education on improved health and suggest that a more physically active lifestyle leads to lower BMI, while sedentary behavior is a consequence of higher BMI.

## Plain language summary

It remains unclear exactly how physical activity, sedentary behavior (usually time spent sitting or lying, often in front of a screen), and obesity influence each other, and what role education plays in this relationship. Here, we use genetic information to study this relationship. We show that if you're more physically active, you're likely to be thinner. If your weight is higher, you tend to spend more time in front of the TV or computer. Additionally, getting more years of education leads to more physical activity, less screen time, and a lower weight later in life. The take-home messages are that being more physically active can prevent obesity; watching more TV is a result but not the cause of obesity; and education stimulates a healthier lifestyle later in life. These findings may help to guide public health messaging around healthy lifestyles.

[1] The Charles Bronfman Institute for Personalized Medicine, Icahn School of Medicine at Mount Sinai, New York, NY, USA. [2] MRC Integrative Epidemiology Unit, University of Bristol and NIHR Bristol Biomedical Research Center, Bristol, UK. [3] Population Health Science, Bristol Medical School, University of Bristol and NIHR Bristol Biomedical Research Center, Bristol, UK. [4] Novo Nordisk Foundation Center for Basic Metabolic Research, Faculty of Health and Medical Sciences, University of Copenhagen, Copenhagen, Denmark. [5] The Beijer Laboratory and Department of Immunology, Genetics and Pathology, Uppsala University and SciLifeLab, Uppsala, Sweden. ✉email: zhe.wang@mssm.edu; marcel.den_hoed@igp.uu.se

Many studies have shown that lower levels of physical activity and more time spent sedentary are associated with higher odds of obesity[1–6]. Other studies suggest that obesity may lead to more sedentary time and lower physical activity levels[3,4]. However, observational and cross-sectional studies can suffer from confounding and reverse causation. Mendelian randomization (MR) studies leverage the special properties of germline genetic variants to strengthen causal inference regarding exposures. MR is now typically implemented within an instrumental variable framework, which allows for a naturally occurring form of randomization that meets the guiding principle of randomized controlled trials, thereby minimizing the influence of reverse causation and confounding[7,8]. MR analyses have shown bidirectional causal effects between more sedentary time and higher BMI, while for physical activity and BMI, results are controversial[9,10]. An earlier study suggested that a higher BMI reduces physical activity, while a more recent study concluded that more physical activity causes a lower BMI but not vice versa[9,10]. Poor instrumental variables for physical activity due to low statistical power in genetic association studies may have limited causal inference between physical activity and BMI. We recently published a large meta-analysis of genome-wide association studies (GWAS) for self-reported leisure time (1) moderate-to-vigorous intensity (MV) physical activity and (2) screen time[11]. Using results from up to 606,820 individuals, we identified 11 loci that are robustly associated with MV physical activity and 88 that are associated with screen time[11]. As a result, we now have stronger instruments for causal inference. However, increasing the number of instruments through ever larger GWAS is not an unalloyed benefit. As the sample size gets larger, it is increasingly likely that variants for upstream traits become genome-wide significant hits for the exposure of interest[7]. So, if years of education influences BMI, eventually single nucleotide variants (SNVs) of primary relevance for education become genome-wide significant hits for BMI. Years of schooling has so far not been taken into consideration when examining the causal relationship of physical activity and sedentary behavior with BMI.

Here, we use genome-wide summary statistics for MV physical activity[11], screen time[11], BMI[12] and years of schooling[13] in individuals of European ancestry as instrumental variables for these traits. We next use a range of univariable and multivariable MR methods[14–19] to assess: (1) the causal relationship between MV physical activity, screen time and BMI; (2) how years of schooling causally affects MV physical activity, screen time and BMI; and (3) how years of schooling affects the causal relationship of MV physical activity and screen time with BMI. The MR methods used include traditional MR methods that use genome-wide significant index SNVs as genetic instrumental variables[15–17] – to facilitate one-to-one comparisons between current and previously published findings, as well as between univariable and multivariable MR methods—as well as the recently introduced Causal Analysis Using Summary Effect estimates (CAUSE)[14] and Latent Heritable Confounder MR (LHC-MR)[19] methods. These utilize full genome-wide summary results, account for both correlated and uncorrelated pleiotropy, and are more robust in some confounding scenarios. The results show that being more physically active can prevent obesity; watching more TV is a result but not the cause of obesity; and education stimulates a healthier lifestyle later in life.

## Methods

**Genetic correlations.** To explore potentially shared genetic architectures, we estimated genetic correlations of MV physical activity and screen time with educational attainment (years of schooling) and obesity related traits (Supplementary Data 1)

using LD score regression implemented in the LD-Hub web resource[20]. To define significance, we applied a Bonferroni correction for the 14 selected phenotypes ($P < 3.5 \times 10^{-3}$).

**Data source for Mendelian Randomization (MR).** We used summary statistics from the largest published meta-analyses of GWAS for MV physical activity, screen time, years of schooling and BMI in individuals of European ancestry (Supplementary Data 2)[11–13]. Results from our recently published physical activity and sedentary behavior GWAS in up to 661,399 European ancestry participants for questionnaire-based, self-reported MV physical activity (more than 20 min per week or not) and screen time (hours per day) were used[11]. We also used GWAS results of educational attainment, assessed as the number of years of schooling completed in 766,345 European ancestry participants[13]. For BMI, we utilized GWAS results from a meta-analysis of the Genetic Investigation of Anthropometric Traits (GIANT Consortium) and the UK Biobank data in 681,275 European participants[12]. We used the GWAS results with the largest sample size to maximize statistical power, acknowledging partial sample overlap. In the presence of weak instruments, sample overlap across traits may bias MR estimates in the direction of the observational association, while no sample overlap between the discovery and outcome would bias estimates towards the null[21]. We limited scope for weak instrument bias by evaluating the strength of instruments as measured by F-statistics[22].

### Univariable two sample MR

*CAUSE and LHC-MR.* We applied the recently published Bayesian-based MR method: CAUSE[14], which accounts for both correlated and uncorrelated pleiotropy and allows overlapping GWAS samples in evaluating bidirectional causal effects between MV physical activity and screen time, years of schooling, and BMI. CAUSE calculates the posterior probabilities of the causal effect and the shared effect, and tests whether the causal model fits the data better than the sharing model. That is, it examines if the association between traits is more likely to be explained by causality than by horizontal pleiotropy. In addition, CAUSE improves the power of MR analyses by using full genome-wide summary results (LD pruned at $r^2 < 0.1$ with $P < 1 \times 10^{-3}$, as recommended for CAUSE)[14]. We also implemented the recently described LHC-MR method that can estimate bidirectional causal effects and confounder effects while accounting for sample overlap[19]. SNVs with minor allele frequency >0.5% were used for LHC-MR analyses.

*Traditional MR.* Genetic instrumental variables for each trait were selected using genome-wide significant ($P < 5 \times 10^{-8}$) index SNVs that were LD clumped ($r^2 > 0.001$ within a 10-Mb window). We followed several steps to evaluate potential causality. As MR results can be severely biased if instrumental SNVs show horizontal pleiotropy and violate the instrumental variable assumptions[16], we prioritized methods that are robust to horizontal pleiotropy when calculating causal estimates. Amongst the methods we prioritized is MR-PRESSO (Pleiotropy RESidual Sum and Outlier)[15], which removes pleiotropy by identifying and discarding influential outlier predictors from the standard inverse variance–weighted (IVW) test[16]. For analyses with no strong evidence of distortion due to pleiotropy (MR-PRESSO Global test $P > 0.05$), we considered other robust methods, for instance fixed- and random-effect IVW, weighted- or simple- median and mode methods. We also conducted Steiger filtering to remove variants likely influenced by reverse causation and used Cook's distance filtering to remove outlying heterogeneous variants as deemed necessary[23]. Outliers identified by Steiger filtering were reported

in Supplementary Data 3. To select the most appropriate approach, we implemented a machine learning framework that predicts the most appropriate model[18].

**Multivariable mendelian randomization**. In the multivariable MR analysis that evaluates the direct effects of each trait, the genetic instrumental variables from two traits were combined. For example, while estimating the direct effect of screen time and years of schooling on BMI, independent loci associated with screen time or years of schooling were pooled together and used as instrumental variables. Conditional F-statistics were calculated to evaluate the strength of the instruments[22]. We set the multiple-testing significance threshold for all MR analyses at 0.008, i.e., Bonferroni correction for the six possible causal effects we are testing: the bidirectional causal effects between screen time or MV physical activity and BMI; years of schooling and BMI; and screen time or MV physical activity and years of schooling (0.05/6 = 0.008). We applied both MR-PRESSO and IVW methods for the multivariable MR analyses, and report MR-PRESSO results when there is evidence of distortion due to pleiotropy, and IVW results otherwise[15].

**Clustering analyses**. For the screen time association signals, we next examined associations with BMI and years of schooling in UK Biobank participants. Among the 88 screen time-associated SNVs ($P < 5 \times 10^{-9}$), 68 SNVs for which associations with screen time, BMI and years of schooling were available to us were used for clustering analyses. We used an agglomerative hierarchical clustering method named 'complete linkage', where each element is its own cluster at the beginning, and two clusters of the shortest distance in between them are sequentially combined into larger clusters until all elements are included in a single cluster[24]. The corresponding $P$-values for, and direction of association with screen time, BMI and years of schooling were used for hierarchical clustering, which yielded five groups of loci. Loci in groups 2 ($n = 8$) and 4 ($n = 6$) are additionally associated with years of schooling; loci in groups 2 and 3 ($n = 9$) with BMI; and loci in group 5 ($n = 42$) are predominantly associated with screen time (Supplementary Fig. 1). For loci in group 5, we cannot exclude the possibility that: (1) variants near but in low LD with lead SNVs ($r^2 < 0.2$) have been associated with obesity traits and/or years of schooling in UK Biobank; (2) lead variants or variants in LD with lead variants have been associated with obesity traits and/or years of schooling in other datasets.

**Reporting summary**. Further information on research design is available in the Nature Portfolio Reporting Summary linked to this article.

## Results
**Univariable MR analyses**. In line with our previous results[11], univariable MR analyses highlight bidirectional causal effects between more MV physical activity and lower BMI, as well as between more screen time and higher BMI (Fig. 1a, b, Supplementary Data 4–5). Moreover, univariable MR analyses reveal bidirectional causal effects between more years of schooling and both more MV physical activity, and less screen time (Fig. 1a, b, Supplementary Data 6–7). More years of schooling also causes a lower BMI, but BMI does not affect years of schooling (Fig. 1, b, Supplementary Data 8).

**Multivariable MR analyses**. In multivariable MR analyses that take BMI into account, the estimated causal effects of MV physical activity and screen time on years of schooling are abolished, such that more years of schooling causes more MV physical activity and less screen time, but not vice versa (e.g., screen time on schooling: Total effect $\beta = -0.49$, $P = 7.8 \times 10^{-24}$; Direct effect $\beta = -0.04$, $P = 0.28$; schooling on screen time: Total effect $\beta = -0.33$, $P = 8.5 \times 10^{-76}$; Direct effect $\beta = -0.40$, $P = 8.8 \times 10^{-55}$; Table 1, Supplementary Data 9, Fig. 1c, d). With that and the aforementioned causal effect of years of schooling on BMI, years of schooling may confound the causal inference between MV physical activity, screen time and BMI. Indeed, when taking years of schooling into account, multivariable MR analyses show that higher BMI has a direct effect on more screen time (Total effect $\beta = 0.16$, $P = 1.4 \times 10^{-74}$; Direct effect $\beta = 0.16$, $P = 1.0 \times 10^{-34}$, Table 1, Fig. 1c), while the effect of more screen time on higher BMI is confounded by years of schooling (Total effect $\beta = 0.40$, $P = 8.4 \times 10^{-14}$; Direct effect $\beta = -0.07$, $P = 0.19$; Table 1, Fig. 1c). In contrast, MV physical activity appears to have a direct effect on BMI (Total effect $\beta = -0.25$, $P = 2.0 \times 10^{-3}$; Direct effect $\beta = -0.20$, $P = 1.8 \times 10^{-6}$)—although precision of the inference may suffer from weak instrument bias (conditional F statistics <10)[22] and results should be interpreted accordingly—while the causal effect of higher BMI on lower MV physical activity appears confounded by years of schooling (Total effect $\beta = -0.10$, $P = 5.8 \times 10^{-12}$; Direct effect $\beta = -0.03$, $P = 0.03$, Supplementary Data 9). It is worth noting that we conclude there is no direct causal effect of BMI on years of schooling (Fig. 1c, d), even though multivariable MR analyses using the IVW method show a significant effect of BMI on years of schooling (Table 1). The conclusion is based on results from the univariable MR analyses using the CAUSE method (Fig. 1a, b, Supplementary Data 8), which is prioritized for its robustness to confounding and pleiotropy.

**Sensitivity analyses using loci that primarily influence screen time**. In line with previous studies[25–27], we found that MV physical activity and screen time are strongly genetically correlated with years of schooling (r 0.62 and −0.58, respectively), as well as with multiple obesity-related traits (r up to −0.33 and 0.41, respectively), suggesting a shared genetic architecture (Supplementary Data 1). In line with high genetic correlations, about one third of the screen time and MV physical activity loci have previously been associated with years of schooling ($n = 11$), obesity-related traits ($n = 11$), or both ($n = 7$)[13,28] (Supplementary Data 10). Therefore, we next perform agglomerative hierarchical clustering using 68 of the 88 screen time-associated SNVs for which associations with years of schooling and BMI are available to us (*Methods*). This yields five groups of screen time-associated loci. Loci in groups 2 ($n = 8$) and 4 ($n = 6$) are additionally associated with years of schooling; loci in groups 2 and 3 ($n = 9$) with BMI; and loci in group 5 ($n = 42$) are predominantly associated with screen time (Supplementary Fig. 1). Based on our MR analyses, we hypothesize that group 5 loci primarily influence screen time directly. We therefore conduct additional sensitivity analyses using group 5 loci only.

When we repeat the two-sample univariable and multivariable MR analyses using only loci predominantly associated with screen time (i.e., group 5 loci, Supplementary Data 11–13), we observe that—similarly to using results from all screen time-associated loci – a 1 SD higher screen time predicts fewer years of schooling, by 0.30 SD ($P = 3.4 \times 10^{-11}$) and increases BMI by 0.28 SD (MR-PRESSO $P = 6.9 \times 10^{-7}$, Supplementary Figure 2a, Supplementary Data 11–12). Results from the sensitivity multivariable MR analyses with group 5 loci only show that the causal effect of higher screen time on higher BMI is confounded by years of schooling (Supplementary Data 13, Supplementary Fig. 2b), supporting the results of our main analyses.

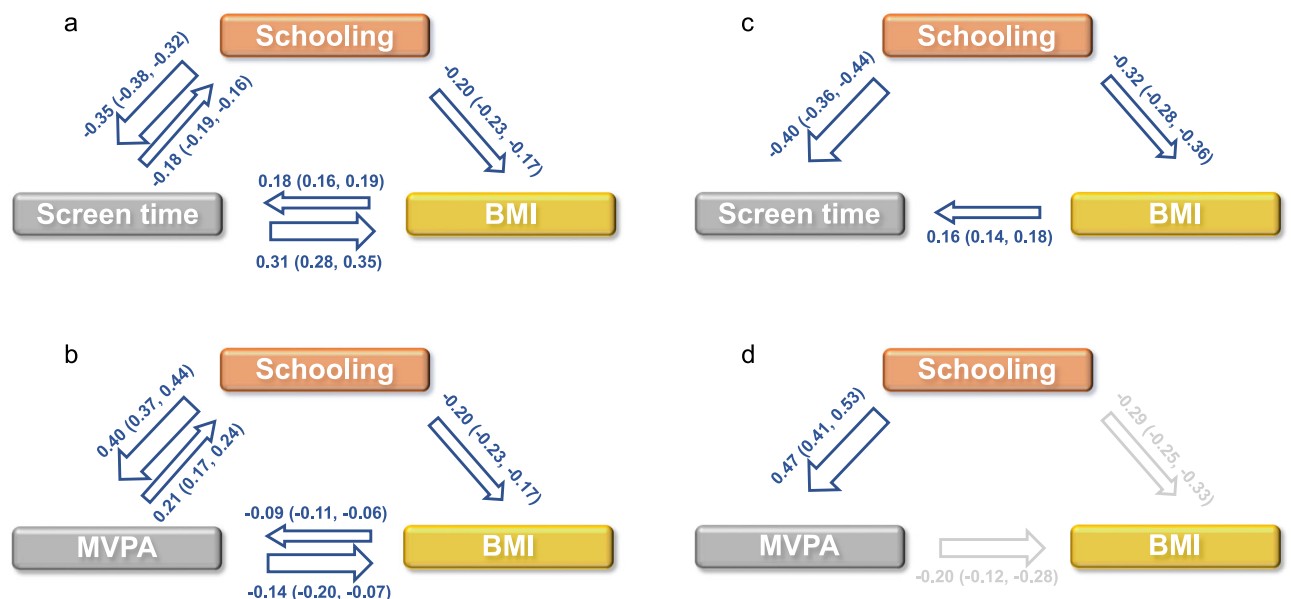

**Fig. 1 Mendelian randomization (MR) analyses between leisure screen time, leisure time moderate-to-vigorous intensity physical activity (MVPA), years of schooling (Schooling) and BMI. a, b** Causal estimates of univariable MR analyses using the CAUSE method for screen time and MVPA. **c, d** Causal estimates of Multivariable MR analyses using the IVW method for screen time and MVPA. Arrows and results in gray indicate that the precision of the effect estimate may suffer from weak instrument bias in the multivariable MR analysis (conditional F statistics <10) and should be interpreted accordingly. Results are shown as effect size (95% CI) for 1 SD change in outcome per 1 SD change in exposure for continuous variables (original units: $kg/m^2$ for BMI, years for Schooling, hours/day for Screen time, and yes/no for MVPA). No direct causal effect of BMI on schooling is shown in (**c, d**) - despite significant effects in multivariable MR analyses - because univariable MR analyses using the CAUSE method show no evidence of a causal effect of BMI on schooling.

**Table 1 Total and direct causal effects estimated from univariable and multivariable Mendelian randomization (MR) analyses for leisure screen time, BMI and years of schooling using the inverse variance weighted (IVW) method.**

| Exposure | Outcome | Total effect | | | Direct effect | | |
|---|---|---|---|---|---|---|---|
| | | beta | SE | *P* | beta | SE | *P* |
| Screen time | BMI | 0.4 | 0.04 | 8.37E-14 | −0.07 | 0.05 | 0.19 |
| BMI | Screen time | 0.16 | 0.01 | 1.35E-74 | 0.16 | 0.01 | 1.00E-34 |
| Schooling | BMI | −0.28 | 0.02 | 4.94E-32 | −0.32 | 0.02 | 1.91E-35 |
| BMI | Schooling | −0.14 | 0.01 | 7.10E-36 | −0.14 | 0.01 | 2.22E-31 |
| Screen time | Schooling | −0.49 | 0.03 | 7.78E-24 | −0.04 | 0.04 | 0.28 |
| Schooling | Screen time | −0.33 | 0.01 | 8.54E-76 | −0.40 | 0.02 | 8.82E-55 |

Schooling: years of schooling; Beta: effect sizes expressed in SD unit changes in outcome per 1 SD increase in exposure. Unit for the exposures and outcomes are SD (original units: $kg/m^2$ for BMI, years for Schooling and hours/day for Screen time); Direct effect: not through the third trait; significant results ($P < 0.008$).
*BMI* body mass index.

## Discussion

In a previous GWAS, a bidirectional causal relationship was reported between accelerometer-assessed overall physical activity and BMI[27]. A lenient threshold was used to select instrumental variables ($P < 5 \times 10^{-6}$), which leaves scope for horizontal pleiotropy, i.e., the selected instrumental variables may influence BMI through mechanisms other than physical activity. Another recent study reported a bidirectional, causal relationship between sedentary time and BMI[9]. MR analyses in our previously published GWAS and the univariable MR analyses in this study show that both MV physical activity and screen time have bidirectional causal relationships with BMI[11], consistent with results from previous studies. However, multivariable MR analyses do not support these bidirectional causal effects (Fig. 1c, d). The main reason for the discrepancy between our and previous studies seems to be that neither of the previous studies have taken years of schooling into account.

A large body of evidence, including randomized controlled trials, suggests that physical activity – particularly MV physical activity—may improve cognitive function and academic achievement[29–31]. Others have shown that a higher attained educational level is associated with more physical activity during leisure time[32]. Additionally, educational attainment is associated with obesity, although the direction of association varies by the country's economic development level: an inverse association is more common in higher-income countries and a positive association is more common in lower-income countries[33]. Using population-based MR approaches to estimate the causal effect between educational attainment and health outcomes may suffer from bias due to dynastic effects[34]. Recent studies applying within-sibship MR have allowed for the estimation of direct causal effects free from such bias[35]. Although these studies have shown bidirectional causal effects between education and BMI, effect estimates were much smaller than those estimated using

population data, especially for BMI on education[36,37]. In line with these findings, we report no direct causal effect of BMI on years of schooling based on the robust CAUSE method, although such an effect is observed when using traditional uni- and multivariable MR methods. This phenomenon further highlights the importance of taking the potential confounder years of schooling into consideration when assessing causal effects between MV physical activity, screen time and BMI[38].

Our multivariable MR results confirm the causal effect of higher BMI on more screen time, but not vice versa. Interestingly, the multivariable MR results for MV physical activity suggest that the effect of higher BMI on lower MV physical activity is confounded by years of schooling. While educational attainment may confound the causal relationship between BMI and physical activity in adulthood, it is less likely to do so in children. The initial study assessing causal effects between childhood obesity and physical activity suggested that higher BMI causes less physical activity and more sedentary time in children at age 11[10]. Another study in younger children showed that 3–8-year-olds with a higher genetic predisposition for obesity spend more time sedentary but are similarly physically active[39], in line with our multivariable MR findings. Similarly to others[40], we observe a direct causal effect of MV physical activity on BMI, although the precision of the effect estimate may suffer from weak instrument bias in the multivariable MR analysis and results should be interpreted accordingly[38]. With the instrumental variable for MV physical activity being stronger in this study than ever before – thanks to a recent doubling of the number of loci robustly associated with MV physical activity[11] – results from earlier MR analyses for MV physical activity should be interpreted with caution.

In summary, our study suggests that more years of schooling will reduce sedentary behavior and BMI, and promote MV physical activity, resulting in improved health. In addition, our results suggest that a more physically active lifestyle results in lower BMI, while reducing sedentary behavior per se – unless it is replaced by MV physical activity – does not affect BMI, but rather is a consequence of higher BMI.

## Data availability

Source data for Fig. 1 can be found in Supplementary Data 4–9. All analyses have been conducted using publicly available data. Our previously published GWAS summary statistics for physical activity and sedentary behavior are available from the GWAS Catalog: https://www.ebi.ac.uk/gwas/publications/36071172. Other GWAS summary statistics used in the analyses described here are freely accessible through the MR-Base platform (https://www.mrbase.org/) and the IEU OpenGWAS database (https://gwas.mrcieu.ac.uk/). All other data and code are available from the corresponding author (ZW), on reasonable request.

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

## Author contributions

Z.W., R.J.F.L. and M.d.H. conceived the study. Z.W. performed data analysis. Z.W., G.D.S., R.J.F.L. and M.d.H. interpreted the results. Z.W. and M.d.H. drafted the paper. All authors reviewed and approved the final version.

## Funding

## Competing interests

The authors declare no competing interests.
