## [Peer Review File · Communications Medicine]

Reviewers' comments:

Reviewer #1 (Remarks to the Author):

Review: Communications Medicine manuscript COMMSMED-23-0279-T
Distilling causality between physical (in)activity and obesity. Wang et al.

This paper uses published GWAS summary statistics to explore the causal relationships between physical activity, screen time, educational attainment and BMI, using several univariable and multivariable MR methods. This is an important topic, and the novelty comes from the use of improved genetic instruments for physical activity, and integration of educational attainment into the causal framework. Once years of schooling are accounted for, the authors conclude that moderate-vigorous physical activity causes lower BMI, and higher BMI is a cause of sedentary behaviour. The paper is generally clearly written, but there are some clarifications needed regarding the presentation of the results.

Table 1 shows the results from the IVW univariable and multivariable MR analysis, whereas Figure 1 shows the results from the CAUSE univariable MR method (part a and b) and multivariable MR (part c and d) – does part c overlap with the Table 1 results?

Why does Table 1 not include the moderate-vigorous physical activity analysis? The Table needs clear headers describing what is presented, including what the beta represents. Is this a SD change in genetically-instrumented outcome per SD change in genetically-instrumented exposure?

Table 1 is only referred to once in the text. The results should be discussed, and also compared with the results in Figure 1, particularly any differences and discrepancies e.g. MVMR effect of BMI on schooling – significant in Table 1 but not shown in Figure 1? Are certain results left out of Figure 1? If so, the reason should be noted in the legend.

Figure 1 - The variables and the causal estimates need to be presented with the relevant units for each trait (what are the units for moderate-vigorous activity), and standard errors or p-values should be shown in the Figure. Are the multivariable MR results from the IVW method?

Extended Data Figure 1. The legend notes darker=lower p value - is there a significance threshold applied?

Reviewer #2 (Remarks to the Author):

Wang et al, conducted a Mendelian randomization investigation for the causal association between physical activity and obesity by using large-scale GWAS summary statistics. They found that more physical activity could cause lower BMI, but not opposite. Years of schooling could cause higher levels of physical activity, less screen time and lower BMI. They then concluded that education has beneficial effect for health, and PA leads to lower BMI. They used datasets from different sources to demonstrate the bidirectional relationship between physical activity and BMI, and the role of education in the above link. A major concern from this reviewer is the weak conceptual advance of this work, with little new knowledge provided for the community or the general readers. Some specific comments:

1) The bidirectional association between physical activity and obesity has been reported previously, such as these listed publications (<https://pubmed.ncbi.nlm.nih.gov/35254260/>)

(<https://pubmed.ncbi.nlm.nih.gov/35990758/>). Meanwhile, the MR association of the education with PA and sedentary behavior have also been reported previously (such as <https://www.nature.com/articles/s41598-021-83801-0>). Therefore, none of these reported methodologies or conclusions are novel given the literature.

2) Analytic framework is also quite confusing. This current study framework is very complicated with too many three-way connections as shown in figure 1. It is far too difficult for a general reader to understand what they want to say or deliver scientifically.

3) The role of schooling in linking MVPA and BMI is still not clear. They did not perform mediation analysis to clarify the mediation role of education/schooling among these relationships, which is feasibility using their current data. Overall, their conclusions are not supported by these results. The current conclusions per se are also a bit confusing, which make one hard to get the key point.

4) Writing of the paper. It is recommended that the authors add some subtitles to the main text to help readers understand the logic of the work.

5) Unique contribution in terms of public health relevance. The current recommendations derived from the manuscript is quite routine. What is the unique contribution compared with the available literature in this aspect?

Reviewers' comments:

Reviewer #1 (Remarks to the Author):

Review: Communications Medicine manuscript COMMSMED-23-0279-T
Distilling causality between physical (in)activity and obesity. Wang et al.

This paper uses published GWAS summary statistics to explore the causal relationships between physical activity, screen time, educational attainment and BMI, using several univariable and multivariable MR methods. This is an important topic, and the novelty comes from the use of improved genetic instruments for physical activity, and integration of educational attainment into the causal framework. Once years of schooling are accounted for, the authors conclude that moderate-vigorous physical activity causes lower BMI, and higher BMI is a cause of sedentary behaviour. The paper is generally clearly written, but there are some clarifications needed regarding the presentation of the results. We thank the reviewer for acknowledging the importance and novelty of our paper, and we appreciate the comments and suggestions to help communicate the results more clearly.

Table 1 shows the results from the IVW univariable and multivariable MR analysis, whereas Figure 1 shows the results from the CAUSE univariable MR method (part a and b) and multivariable MR (part c and d) – does part c overlap with the Table 1 results?

Yes, the reviewer correctly concludes that figure 1 part c partially overlaps with the table 1 results. However, as the reviewer points out further down, there are some differences between figure 1c and table 1. Most importantly, the significant multivariable MR effect of BMI on years of schooling using the IVW method is shown in Table 1, but not in Figure 1c, because univariable MR analyses using the CAUSE method do not show a direct causal effect of BMI on years of schooling (shown in Figure 1a and in Supp Table 6). We prioritize results from the CAUSE method over those obtained using other methods – including IVW – because CAUSE is robust to confounding and pleiotropy. Since we conclude from univariable MR analyses that there is no causal effect of BMI on years of schooling, we decided against reporting multivariable MR results for BMI on years of schooling in Figure 1c. We still include the univariable MR and multivariable MR results for BMI on years of schooling in Table 1 for completeness.

We now clarify this in the text as follows: “It is worth noting that we conclude there is no direct causal effect of BMI on years of schooling (**Figure 1c-d**), even though multivariable MR analyses using the IVW method show a significant effect of BMI on years of schooling (**Table 1**). The conclusion is based on results from the univariable MR analyses using the CAUSE method (**Figure 1a-b, Supp Table 6**), which is prioritized for its robustness to confounding and pleiotropy.” (Page 5, lines 85-89).

We also added the following note to the legend of Figure 1, as the reviewer suggests: “No direct causal effect of BMI on schooling is shown in c-d – despite significant effects in multivariable MR analyses – because univariable MR analyses using the CAUSE method show no evidence of a causal effect of BMI on schooling.”

Why does Table 1 not include the moderate-vigorous physical activity analysis? The Table needs clear headers describing what is presented, including what the beta represents. Is this a SD change in genetically-instrumented outcome per SD change in genetically-instrumented exposure?

Results for MV physical activity analyses are shown in Supplementary table 7, rather than in Table 1. We consider the multivariable MR results for MV physical activity supplementary, because they may suffer from weak instrument bias due to conditional F statistics <10.

We thank the reviewer for pointing out that further clarification is desirable. We have now added the three traits of interest in the table header, and added the following information to the footnote: “Beta: effect sizes expressed in SD unit changes in outcome per 1 SD increase in exposure.” In addition to the original footer stating that: “Unit for the exposures and outcomes are SD (original units: kg/m² for BMI, years for Schooling and hours/day for Screen time); ...”.

Table 1 is only referred to once in the text. The results should be discussed, and also compared with the results in Figure 1, particularly any differences and discrepancies e.g. MVMR effect of BMI on schooling – significant in Table 1 but not shown in Figure 1? Are certain results left out of Figure 1? If so, **the reason should be noted in the legend.**

Thank you for this great suggestion. We have now revised the main text on pages 4-5 to add more discussion for the Table 1 results. Please also see our earlier response regarding the differences and discrepancies between Table 1 and Figure 1.

Figure 1 - The variables and the causal estimates need to be presented with the relevant units for each trait (what are the units for moderate-vigorous activity), and standard errors or p-values should be shown in the Figure. Are the multivariable MR results from the IVW method?

The reviewer is correct, the figure should include information on the variance in effect estimates. We now show effect size and 95% CI in Figure 1 and effect size \pm SE in Extended Data Figure 2. We reported 95% CI rather than standard errors in Figure 1 because the CAUSE method reports the effect estimate gamma – which is a posterior median that can be taken as a point estimate of the causal effect – and its 95% confidence interval (<https://pubmed.ncbi.nlm.nih.gov/32451458/>). We also added information in the legend to illustrate the relevant units. That is: (Figure 1) “Results are shown as effect size (95% CI) for 1 SD change in outcome per 1 SD change in exposure for continuous variables (original units: kg/m² for BMI, years for Schooling, hours/day for Screen time, and yes/no for MVPA).”; (Extended Data Figure 2) “Results are shown as beta \pm SE: expressed in SD unit changes in outcome per 1 SD increase in exposure (original units: kg/m² for BMI, years for Schooling and hours/day for Screen time).”. Multivariable MR results are from the IVW method, which has now been added.

Of note: leisure time MV physical activity was analyzed as a dichotomous trait (yes/no) in the GWAS. While most studies used ≥ 20 min/week of any leisure moderate/vigorous activity to define participation in MV physical activity, the exact activities considered when calculating the MV physical activity trait vary amongst the 51 studies included in the meta-analysis (details can be found in the GWAS paper <https://www.nature.com/articles/s41588-022-01165-1>).

Extended Data Figure 1. The legend notes darker=lower p value - is there a significance threshold applied?

Thank you for flagging this. Yes, we only included loci that are significantly associated with leisure screen time ($P < 5 \times 10^{-9}$). We now clarify this in the main text methods section: “Among the 88 screen time-associated SNVs ($P < 5 \times 10^{-9}$), 68 SNVs for which associations with screen time, BMI and years of schooling were available to us were used for clustering analyses.” (Page 10, lines 236-238). We also updated the extended data figure 1 title: “Extended Data Figure 1. Agglomerative hierarchical clustering identifies five clusters of leisure screen time-associated loci ($P < 5 \times 10^{-9}$) based on associations with BMI and years of schooling.”

Reviewer #2 (Remarks to the Author):

Wang et al, conducted a Mendelian randomization investigation for the causal association between physical activity and obesity by using large-scale GWAS summary statistics. They found that more physical activity could cause lower BMI, but not opposite. Years of schooling could cause higher levels of physical activity, less screen time and lower BMI. They then concluded that education has beneficial effect for health, and PA leads to lower BMI. They used datasets from different sources to demonstrate the bidirectional relationship between physical activity and BMI, and the role of education in the above link. A major concern from this reviewer is the weak conceptual advance of this work, with little new knowledge provided for the community or the general readers. Some specific comments:

1) The bidirectional association between physical activity and obesity has been reported previously, such as these listed publications (<https://pubmed.ncbi.nlm.nih.gov/35254260/>) (<https://pubmed.ncbi.nlm.nih.gov/35990758/>). Meanwhile, the MR association of the education with PA and sedentary behavior have also been reported previously (such as <https://www.nature.com/articles/s41598-021-83801-0>). Therefore, none of these reported methodologies or conclusions are novel given the literature.

We agree with the reviewer that others have examined the causal relationship between physical activity and BMI before us, as pointed out in the papers highlighted by the reviewer that we refer to in our manuscript. Equally, others have reported a causal effect of education on odds of obesity. The novelty of our study comes from: 1) the use of improved genetic instruments for MV physical activity and sedentary behavior, facilitated by our recently published meta-analysis of GWAS results across 51 cohort studies and UK Biobank; 2) integration of educational attainment into the causal framework; and 3) implementation of improved MR methods and analyses to distill the intricate causal triangulation between these traits. Thanks to these factors combined, our conclusions on the causal relationship between physical (in)activity and BMI are more robust than those drawn in earlier studies, and they are different. We conclude – for the first time – that a more physically active lifestyle results in lower BMI, while reducing sedentary behavior per se (unless it is replaced by MV physical activity) does not affect BMI, but rather is a consequence of higher BMI. We try to emphasize this in our manuscript, e.g.:

a. “MR analyses have shown bidirectional causal effects between more sedentary time and higher BMI, while for physical activity and BMI, results were controversial^{9,10}. An earlier study suggested that a higher BMI reduces physical activity, while a more recent study concluded that more physical activity causes a lower BMI but not *vice versa*^{9,10}” (Page 3, lines 27-30);

b. “Similarly to others³⁶, we observe a direct causal effect of MV physical activity on BMI, although the precision of the effect estimate may suffer from weak instrument bias in the multivariable MR analysis³⁴. With the instrumental variable for MV physical activity being stronger in this study than ever before – thanks to a recent doubling of the number of loci robustly associated with MV physical activity¹¹ – results from earlier MR analyses for MV physical activity should be interpreted with caution.” (Page 7, lines 157-162);

c. “Additionally, educational attainment is associated with obesity, although the direction of association varies by the country's economic development level: an inverse association is more common in higher-income countries and a positive association is more common in lower-income countries²⁹. Using population-based MR approaches to estimate the causal effect between educational attainment and health outcomes may suffer from bias due to dynastic effects³⁰. Recent studies applying within-sibship MR have allowed for the estimation of direct causal effects free from such bias³¹. Although these studies

have shown bidirectional causal effects between education and BMI, effect estimates were much smaller than those estimated using population data, especially for BMI on education^{32,33}." (Pages 6-7, lines 127-143).

2) Analytic framework is also quite confusing. This current study framework is very complicated with too many three-way connections as shown in figure 1. It is far too difficult for a general reader to understand what they want to say or deliver scientifically.

We appreciate the reviewer's concern and have further explained the rationale for our MR analyses, which aim to disentangle the complicated triangular relationships between physical (in)activity, years of schooling and BMI (Pages 4-5).

3) The role of schooling in linking MVPA and BMI is still not clear. They did not perform mediation analysis to clarify the mediation role of education/schooling among these relationships, which is feasibility using their current data. Overall, their conclusions are not supported by these results. The current conclusions per se are also a bit confusing, which make one hard to get the key point.

We thank the reviewer for further pointing out what is not clear in our results section. We agree that our current data are able to clarify the role of education on the causal effect between MVPA and BMI, which is one of our primary goals and the novelty of this paper. In order to investigate the role of years of schooling on the bidirectional causal effect between MVPA and BMI, we performed multivariable MR analyses. These can be used to estimate mediating effects, or to adjust for potential confounding. Our multivariable MR analyses show that years of schooling has a direct effect on BMI and on MVPA, while BMI and MVPA do not affect years of schooling. Therefore, in the causal relationships between MVPA and BMI, years of schooling may work as a confounder, but not as a mediator in the causal relation between MVPA and BMI. Indeed, multivariable MR analyses that take years of schooling into account reveal that more MVPA is causally associated with a lower BMI, while the causal effect of higher BMI on lower MVPA appears confounded by years of schooling (Total effect $\beta = -0.10$, $P = 5.8 \cdot 10^{-12}$; Direct effect $\beta = -0.03$, $P = 0.03$).

We now further clarify our analyses and results (Pages 4-5).

4) Writing of the paper. It is recommended that the authors add some subtitles to the main text to help readers understand the logic of the work.

We agree that headers and subtitles increase readability and are helpful to guide the author through the manuscript. Our manuscript was originally written as a short communication, and subtitles are not used in those. We have now added headers and subtitles in the main text, as the reviewer suggests.

5) Unique contribution in terms of public health relevance. The current recommendations derived from the manuscript is quite routine. What is the unique contribution compared with the available literature in this aspect?

As mentioned in response to comment 1: we use stronger instrumental variables than were available to date, as well as a range of state-of-the-art MR methods, to explore – for the first time – the triangular causal interplay between education, physical (in)activity and BMI. This allows us to draw more robust conclusions than others have been able to draw before us about the bidirectional causal relation between physical (in)activity and BMI. These new conclusions have implications for public health recommendations. That is: weight loss through physical activity can likely be achieved by spending more leisure time in MVPA, but not by reducing leisure screen time per se. These conclusions are drawn at the end of the abstract and main paper.

Reviewers' comments:

Reviewer #1 (Remarks to the Author):

Thank you to the authors for their detailed response and clarifications. I have some further questions based on the response.

1. There is a significant effect of BMI on schooling in the IVW-MVMR as shown in Table 1, but this is not shown in Figure 1 (as not significant in the univariable CAUSE method). This seems to be mixing up methods and findings from 2 different approaches. If the IVW-MVMR is the best available method for multivariable MR then these results should be presented in full, but any caveats can be discussed in the text (which could potentially affect all the MVMR results?). Also, is it possible that even if there is no significant univariable MR effect, there may be still be a valid direct effect in a multivariable framework?

2. CAUSE method vs IVW method for univariable MR. The authors consider the CAUSE method to be preferable, but also present the IVW univariable method in Table 1. The 2 methods give conflicting results and these differences should be discussed. Given this is a short article, is it better to focus on one univariable method, with the other as sensitivity, unless a method comparison is being made and discussed?

3. If the IVW-MVMR method is considered unreliable due to weak instrument bias for the conditional F-statistic for MV physical activity, why is part d shown in Figure 1? The abstract (page 2 line 11-12) states that "more MV physical activity causes a lower BMI", but this part of Figure 1 is greyed out. Also, given the weak instrument, are the results for more schooling causing higher MV physical activity reliable?

Minor points:

4. The actual SD values for these variables in the datasets involved could be noted somewhere eg supplement or Table footnote/Figure legend.

5. It could be noted in the main text eg p 3 line 46-51, that this study is conducted in European ancestry populations.

Reviewer #2 (Remarks to the Author):

the authors have addressed my prior comments.

Reviewers' comments:

Reviewer #1 (Remarks to the Author):

Thank you to the authors for their detailed response and clarifications. I have some further questions based on the response.

1. There is a significant effect of BMI on schooling in the IVW-MVMR as shown in Table 1, but this is not shown in Figure 1 (as not significant in the univariable CAUSE method). This seems to be mixing up methods and findings from 2 different approaches.

If the IVW-MVMR is the best available method for multivariable MR then these results should be presented in full, but any caveats can be discussed in the text (which could potentially affect all the MVMR results?).

Also, is it possible that even if there is no significant univariable MR effect, there may be still be a valid direct effect in a multivariable framework?

We thank the reviewer for raising this point of unclarity. Rather than mixing up the results of different methods, we have carefully and deliberately presented them in the way we have to stress important points and avoid (rather than cause) confusion. In the main results section on UVMR results (lines 61-66), we only provide results of the CAUSE method that we consider the most robust, and we only refer to **Figure 1**, which summarizes – as far as UVMR results go – the CAUSE results that our conclusions are based on. The aim of only providing IVW results in **Table 1** (not referred to in the UVMR part of the results section) is to contrast results of the UVMR and MVMR analyses in a fair manner (see the response to comment 2 below).

The UVMR and MVMR approaches serve different purposes. While the UVMR approach assesses the total causal effect of an exposure on an outcome, which might be affected by another related exposure, the MVMR approach can be used as the equivalent of a mediation analysis within the MR framework, i.e., to decompose the effects of an exposure on an outcome that are direct vs. those that act via potentially related exposures. In the absence of a total effect of one trait on another (based on the CAUSE UVMR result), there is no point in discussing if that non-existing effect is mediated or confounded by another trait (using IVW MVMR). We elaborate on this rationale in lines 85-90.

2. CAUSE method vs IVW method for univariable MR. The authors consider the CAUSE method to be preferable, but also present the IVW univariable method in Table 1. The 2 methods give conflicting results and these differences should be discussed. Given this is a short article, is it better to focus on one univariable method, with the other as sensitivity, unless a method comparison is being made and discussed?

The reviewer is correct in concluding that within the UVMR approach, we consider the CAUSE method to provide more robust estimates of causal effects than the IVW method (see lines 57-58, 89-90).

Importantly, within the UVMR approach, we already focus on the CAUSE results as the main result (lines 61-66). However, since there is no CAUSE method within the MVMR framework, a comparison of CAUSE UVMR with IVW MVMR in Table 1 would not just contrast UVMR and MVMR, but also CAUSE and IVW. From such a comparison, one could not conclude whether differences are driven by the method (IVW vs. CAUSE) or the approach (UVMR vs. MVMR). Since the IVW method is still a mainstream method in MR studies, and since this method has been used in previously published studies on this topic – in which the conclusions were different to ours, as we discussed – we think it makes sense to present results from the most robust UVMR method (CAUSE) in the main figure; from the IVW UVMR vs. IVW MVMR in Table 1, and from the various UVMR methods in supplementary tables. We have now expanded on our rationale for providing results from multiple UVMR methods at the end of the introduction section (lines

53-54). There are of course multiple ways in which the results can be presented, but we hope the reviewer can appreciate the rationale for the approach we took.

3. If the IVW-MVMR method is considered unreliable due to weak instrument bias for the conditional F-statistic for MV physical activity, why is part d shown in Figure 1? The abstract (page 2 line 11-12) states that “more MV physical activity causes a lower BMI”, but this part of Figure 1 is greyed out.

Also, given the weak instrument, are the results for more schooling causing higher MV physical activity reliable?

Thank you for pointing out a potential source of unclarity. As we state in the discussion section: “the **precision** of the effect estimate **may** suffer from weak instrument bias in the multivariable MR analysis” (line 149-151). Importantly, this does **not** imply that the **results** are considered unreliable. In fact, these results are more reliable than the causal effect estimates published previously for MV physical activity, thanks to a doubling of the number of MV physical activity-associated loci in our recently published meta-analysis of GWAS (as discussed in lines 151-154).

For clarity, we have now updated the legend of Figure 1 to say the same thing, and we have added the statement on precision of the effect estimate to the results section as well (lines 82-83). We have added these statements to make sure readers are aware of the limitations of this analysis caused by the weak instrument for MVPA, which applies even more so to causal inference for MVPA on BMI in earlier studies.

The precision of the results for more years of schooling causing higher MV physical activity are not affected by a weak instrument, because – as shown in Supplementary Table 7 – the conditional F statistic for the MVMR test of exposures years of schooling and BMI on outcome MV physical activity is 15.04, which is greater than the threshold of 10, suggesting the instrument is strong enough.

Minor points:

4. The actual SD values for these variables in the datasets involved could be noted somewhere eg supplement or Table footnote/Figure legend.

We thank the reviewer for this suggestion and have now added a new Supplementary Table 12 that provides this information.

5. It could be noted in the main text eg p 3 line 46-51, that this study is conducted in European ancestry populations.

We thank the reviewer for suggesting to add this information to the main paper as well. We have now added it to the main text (line 47) and to the new Supplementary Table 12. This information was already included in the methods section (lines 169-170).

REVIEWERS' COMMENTS:

Reviewer #1 (Remarks to the Author):

Thank you to the authors for their response and clarifications. The authors have satisfactorily responded to my questions and added further explanation to the manuscript. I have no further comments.